# Large-area, periodic, and tunable intrinsic pseudo-magnetic fields in low-angle twisted bilayer graphene

Haohao Shi[1,2,6], Zhen Zhan [3,6], Zhikai Qi[4], Kaixiang Huang [3], Edo van Veen[5], Jose Ángel Silva-Guillén [3], Runxiao Zhang[1,2], Pengju Li[1,2], Kun Xie[1,2], Hengxing Ji [4], Mikhail I. Katsnelson[5], Shengjun Yuan [3]*, Shengyong Qin [1,2]* & Zhenyu Zhang [1]

A properly strained graphene monolayer or bilayer is expected to harbour periodic pseudo-magnetic fields with high symmetry, yet to date, a convincing demonstration of such pseudo-magnetic fields has been lacking, especially for bilayer graphene. Here, we report a definitive experimental proof for the existence of large-area, periodic pseudo-magnetic fields, as manifested by vortex lattices in commensurability with the moiré patterns of low-angle twisted bilayer graphene. The pseudo-magnetic fields are strong enough to confine the massive Dirac electrons into circularly localized pseudo-Landau levels, as observed by scanning tunneling microscopy/spectroscopy, and also corroborated by tight-binding calculations. We further demonstrate that the geometry, amplitude, and periodicity of the pseudo-magnetic fields can be fine-tuned by both the rotation angle and heterostrain. Collectively, the present study substantially enriches twisted bilayer graphene as a powerful enabling platform for exploration of new and exotic physical phenomena, including quantum valley Hall effects and quantum anomalous Hall effects.

[1] International Centre for Quantum Design of Functional Materials (ICQD), Hefei National Laboratory for Physical Sciences at the Microscale (HFNL), and Synergetic Innovation Center of Quantum Information and Quantum Physics, University of Science and Technology of China, Hefei 230026, China. [2] CAS Key Laboratory of Strongly-Coupled Quantum Matter Physics, Department of Physics, University of Science and Technology of China, Hefei 230026, China. [3] Key Laboratory of Artificial Micro- and Nano-structures of Ministry of Education and School of Physics and Technology, Wuhan University, Wuhan 430072, China. [4] Hefei National Laboratory for Physical Sciences at the Microscale, Department of Applied Chemistry, CAS Key Laboratory of Materials for Energy Conversion, iChEM (Collaborative Innovation Center of Chemistry for Energy Materials), University of Science and Technology of China, Hefei 230026, China. [5] Institute for Molecules and Materials, Radboud University, Heyendaalseweg, 135, 6525 AJ, Nijmegen, The Netherlands. [6]These authors contributed equally: Haohao Shi, Zhen Zhan. *email: s.yuan@whu.edu.cn; syqin@ustc.edu.cn

Strain in graphene can be served as an effective tuning parameter in tailoring its exotic properties reflected by different degrees of freedom such as spin and valley[1–6]. In particular, it has been predicted that a non-uniform strain distribution imposed on graphene arising from either external stress[1,2] or interfacial coupling[3,4] can induce a strong gauge potential that in turn can act as a pseudo-magnetic field (PMF) on the Dirac electrons. Indeed, it has been demonstrated experimentally that locally strained graphene in the form of nanobubbles can induce PMFs, as manifested by the emergent pseudo-Landau levels observed using the scanning tunneling microscopy/spectroscopy (STM/S)[5,6]. More recently, existence of large-area, spatially distributed PMFs has also been reported for single-layer graphene on a black phosphorus substrate, which provides strain textures due to the mismatched symmetries and lattice constants of the heterostructure[7]. Separately, intensive theoretical and experimental studies have demonstrated that low-angle twisted bilayer graphene (TBG) can exhibit a variety of exotic phenomena, including superconductivity and quantum phase transitions[8–19]. In addition, it has been reported that a small uniaxial heterostrain in low-angle TBG can suppress Dirac cones and lead to the emergence of a zero energy resonance[20]. In practice, low-angle TBG itself possesses large-scale, natural, and periodic strains due to the incommensurate moiré superstructures[3,4,21]. Such strain has been theoretically studied to significantly affect the electronic properties of the large-scale moiré pattern[22,23]. However, yet to date, experimental explorations of such natural and intrinsic strain effects arising from lattice relaxations in the strongly coupled homostructural system, especially the pseudo-magnetic behaviors, are still lacking[22,24,25].

In this letter, by combining STM/S experiments and large-scale tight-binding calculations, we study the electronic properties of TBG with extremely small twist angles ($\theta < 1°$). We directly detect pseudo-Landau levels generated by the intrinsic PMFs, which originate from the interplay between interlayer interactions and in-plane strain fields. The PMFs form a ring-structured vortex lattice associated with the moiré pattern, and are periodically distributed over the whole sample. The vortex lattice, in which all angular incommensurability is focused at specific singularity points, is chiral for the charge carriers at different valleys in

$k$-space, and can be further tuned, together with the PMFs, by the twist angle and uniaxial heterostrain in the bilayer system. Indeed, the existence of periodic PMFs in such hexagon-structured system provides a possible platform to further explore quantum anomalous Hall effects if a tiny external magnetic field is applied to break the time-reversal symmetry[26].

## Results

**Pseudo-magnetic fields with twist angle $\theta = 0.48°$.** Our samples were prepared by chemical vapor deposition on Cu–Ni alloyed substrates. Cu–Ni alloyed substrates have been widely used to grow bilayer graphene[27]. High quality bilayer graphene samples with multiple domains were obtained after appropriate annealing processes (Supplementary Note 1). Figure 1a shows an STM topography image with hexagonal symmetry moiré patterns. The atomic structure of the moiré pattern can be determined explicitly by the topography image[20,28,29]. Here, we follow the same method to identify the twist angle in Fig. 1a, which gives $\theta = 0.48°$ (Supplementary Note 5).

Next, we characterize the electronic properties of the TBG samples by the dI/dV tunneling spectra. Generally, dI/dV conductance is proportional to the local density of states in the vicinity of the STM tip position. To rule out the possible tip effects on the measured dI/dV conductance, we have performed dI/dV measurements on TBG with large twist angles. The obtained spectra show typical angle-dependent van Hove singularities which are highly consistent with previous reports[30] (Supplementary Note 2). Figure 1b shows dI/dV spectra taken in both AA and AB stacking regions. The AA region spectra show a series of pronounced peaks (indicated by numbers) with nearly equal energy spacing (Supplementary Note 3). Surprisingly, these peaks are typically different from those of TBG in the absence of external magnetic fields[10,20,30,31], but rather similar to the Landau levels of a two dimensional electron gas in graphene under strong external magnetic fields[32,33]. In stark contrast, these resonances are hardly found in the AB region. For extremely low-angle TBG, the charge density in the AB region resembles that of ideal Bernal-stacking bilayer graphene[4].

To interpret our experimental observations, we have performed theoretical calculations of the superstructure shown in Fig. 1a.

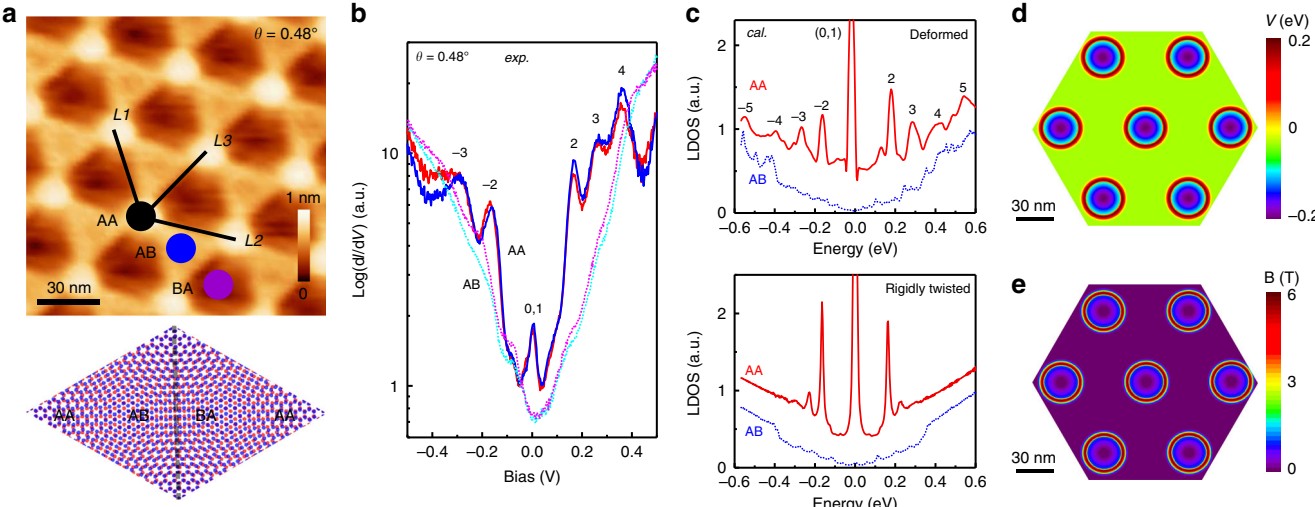

**Fig. 1 Electronic properties of TBG with twist angle $\theta = 0.48°$. a** STM topography image (100 nm × 100 nm) with hexagonal moiré pattern (top panel) and the atomic model of AA, AB, and BA stacking regions (bottom panel). The three moiré wavelengths are: $L1 \approx L2 \approx L3 \approx 29.6$ nm. Sample bias $V = 100$ mV, tunneling current $I_t = 1.0$ nA. **b** Logarithmic dI/dV spectra measured at AA and AB regions. The two solid (dashed) lines were taken at different AA (AB) regions to show reproducibility (curves are vertically shifted for clarity). **c** Calculated local density of states in the AA and AB regions for deformed (upper panel) and rigidly twisted (bottom panel) cases. **d, e** Calculated local potential $V$ and pseudo-magnetic fields **B**, respectively.

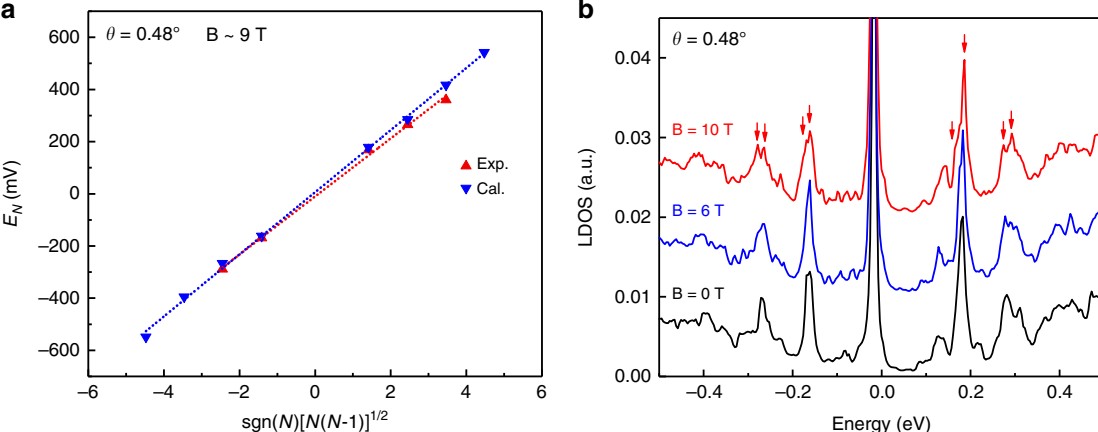

**Fig. 2 Pseudo-Landau levels in the deformed twisted bilayer graphene with $\theta = 0.48°$. a** Linear fit of the equation $E_N \propto \sqrt{N(N-1)}$ and the obtained pseudo magnetic fields is about 9 T. **b** Calculated LDOS at AA region under the external magnetic fields, in which we can confirm the splittings of the pseudo-Landau level due to the break of the valley degeneracy.

The as shown moiré pattern unit cell contains 57964 atoms that is too large even for state-of-the-art first-principles methods. Therefore, we adopted a widely used full tight-binding model of bilayer graphene with an angle-dependent Hamiltonian[9,20]. Figure 1c shows the calculated local density of states by utilizing the Lanczos recursion method in real space[34]. Considering the lattice deformation of the bilayers (upper panel), the result reproduces the energy peaks that appear in the measured spectra (Fig. 1b) whereas the appearance of high energy resonances are dramatically different in the rigidly twisted one (bottom panel). The implementation of the lattice distortion in a rigidly twisted bilayer graphene will be discussed later. As a side issue, our calculations (Fig. 1c) also reproduce the observed resonance peak at the Fermi energy. Recent investigations of such system have provided experimental evidence of such strong electronic correlations near the magic angle TBG by means of STM/S[16–19].

For bilayer graphene under external magnetic fields, the massive Dirac electrons are quantized with energies $E_N = \pm\hbar\omega_c\sqrt{N(N-1)}$, $\omega_c = eB/m^*$, $N = 0, 1, 2,\ldots$, where $\hbar$ is Planck's constant and $m^*$ is the effective mass of electrons[24,33]. Here, the resonant peak positions are linear with $\sqrt{N(N-1)}$, indicating the feature of pseudo-Landau levels in bilayer graphene. For the TBG with twist angle of 0.48°, the fitted PMF value is about 9 T, as shown in Fig. 2a. In addition, the PMF in low-angle TBG does not break the time-reversal symmetry and we further calculated the pseudo-Landau levels under external magnetic fields (Fig. 2b). An interesting observation is that, under the strong external magnetic fields, the pseudo-Landau levels are splitted into two peaks with an energy gap of about 15 meV due to the breaking of valley degeneracy. Similar phenomena have been reported in earlier studies of strained monolayer graphene[35] except that in the present study, such valley splitting is not as large as theoretically expected. The suppression can be understood, because the theoretically expected valley splitting is valid only for well-defined K and K′ valleys, where the PMF is uniform. While in our systems the observed PMF has drastically varying ring-like structures in real space. As a result, the two valleys are not well defined; therefore, the theoretically expected valley splitting value is no longer applicable. Instead, the valley splitting observed in this study is extrapolated from the regions where the pseudo-magnetic field only varies mildly in real space.

**Theoretical calculations.** To illustrate the delicate physics involved in reaching the agreement between the theory and experiment shown in Fig. 1, we note that, in graphene-based van der Waals structures, compelling evidences have revealed that the lattice relaxation effects are crucial in determining their electronic properties[3,4,15,29]. Especially, in low-angle TBG, the super-structure undergoes lattice distortions and structural transformations, which can be described as a lattice deformation of the moiré superlattice due to the interplay between the interlayer interaction and the in-plane strain fields. That is, on one hand, the in-plane forces move atoms to maximize the area of AB/BA stacking domains, which has the minimum binding energy. On the other hand, the strain induced by the atomic displacements hinders such in-plane atomic rearrangement[3,4]. By fitting the atomic structures obtained from molecular dynamics simulations of TBG in refs. [3,4], the in-plane displacement $\Delta d$ and out-of-plane displacement $\Delta z$ of individual atoms can be approximately expressed as:

$$\begin{aligned}\Delta d(r) &= \Delta D \cdot \sin\left(\frac{\pi r}{2l_D}\right)[1 - \Theta(r - l_D)] \\ &\quad + \Delta D \cdot e^{[-(r-l_D)^2/\sigma_D]}\Theta(r - l_D), \\ \Delta z(r) &= \pm \Delta Z \cdot e^{(-r^2/\sigma_Z)}[1 - \Theta(r - l_Z)],\end{aligned}$$

(1)

where $r$ is the distance between the individual atoms and the nearest AA point in the x–y plane, $\Theta$ is the Heaviside step function, $\Delta D$ is the maximum in-plane displacement and $\Delta Z$ is the maximum out-of-plane displacement that are dependent on the rotation angle[4]. $l_D$, $\sigma_D$, $l_Z$, $\sigma_Z$ are constants.

The deformed samples are constructed using the following procedure: First, the rigidly twisted bilayer graphene is constructed according to the commensurability (Supplementary Note 5) and the position of each atom is well known. Next, we modify the in-plane and out-of-plane positions of atoms according to the position-dependent expression in Eq. (1) (Supplementary Note 6). For instance, for the in-plane deformation, an individual atom moves $\Delta d$ towards the nearest AA point. In the out-of-plane deformation, each individual atom has a displacement in the z direction with + sign for the top layer and − sign for the bottom layer. The maps of $|\Delta d|$ and $|\Delta z|$ for TBG with $\theta = 0.48°$ are plotted in Fig. 3c, d, respectively. The movement of the atoms have the same tendency as that obtained from molecular dynamics simulations in reference[4]. In this part, we assume that the deformed sample keeps the period of the rigid TBG. In the rigid sample, we use the AA point as the rotation center. The moiré supperlattice has point group $D_3$ generated by a three-fold rotation about the axis formed by the AA point ($C_3$) and a two-fold

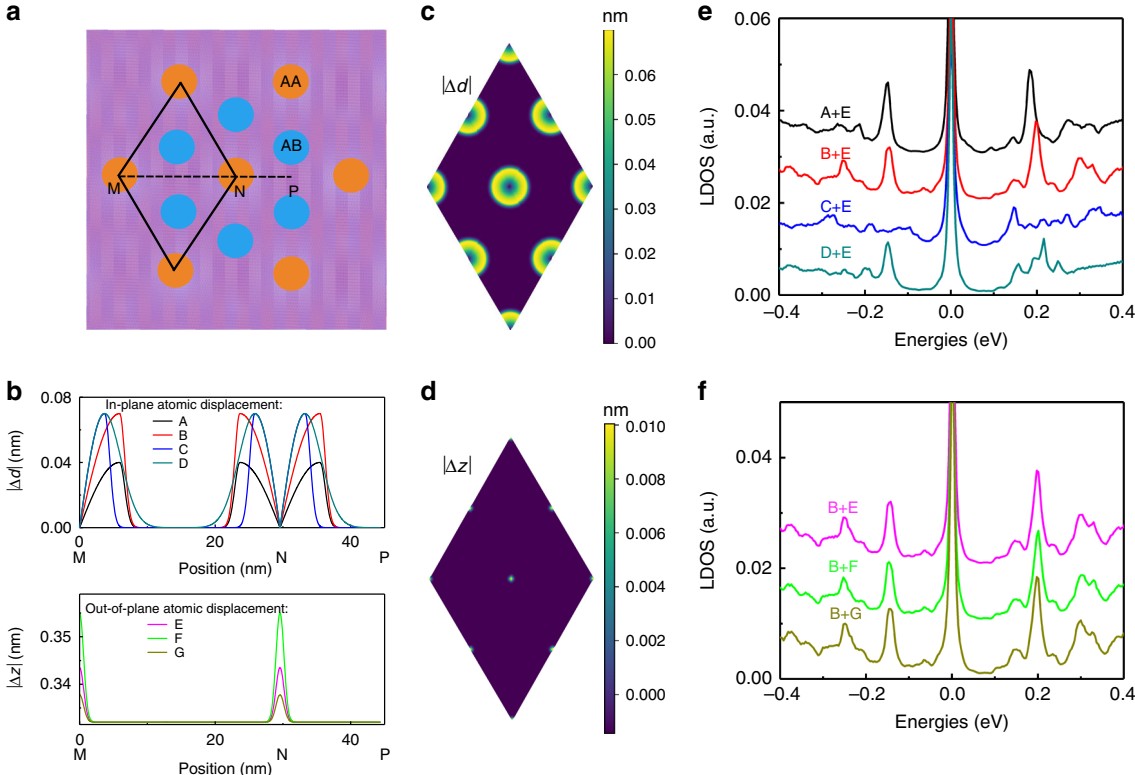

**Fig. 3 Calculated results of the distorted TBG structure with $\theta = 0.48°$. a** Schematic model of the moiré pattern. **b** Absolute magnitude of different in-plane atomic displacements (top panel) and out-of-plane displacements for the deformed system along the path MNP defined in **a**. For the in-plane displacements, the parameters $(\Delta D, l_D, \sigma_D)$ are: A (0.04 nm, 5.91 nm, 1.14), B (0.07 nm, 5.91 nm, 1.14), C (0.07 nm, 3.70 nm, 1.14), D (0.07 nm, 3.70 nm, 11.44). For the out-of-plane displacements, the parameters $(\Delta Z, l_Z, \sigma_Z)$ are: E (0.0115 nm, 4.93 nm, 0.7), F (0.023 nm, 4.93 nm, 0.7), G (0.0058 nm, 4.93 nm, 0.7). **c, d** Maps of the absolute magnitude of the in-plane atomic displacement $|\Delta d|$ and out-of-plane displacement $\Delta z$ upon deformation of rigidly TBG calculated from Eq. (1), respectively. **e, f** Local density of states in the AA region of deformed systems with different in-plane and out-of-plane displacements, respectively.

rotation about the axis perpendicular to the AA point ($C_2'$). The deformed sample maintains the $D_3$ point group symmetry[36,37]. For another sample with uniaxial heterostrain, the supperlattice for both rigidly twisted and deformed cases lose the $D_3$ symmetry. In Eq. (1), the parameters $\Delta D$, $l_D$, $\sigma_D$, $\Delta Z$, $l_Z$, and $\sigma_Z$ determine the profile of the deformed structure and are fitting parameters to generate proper structures with properties matching the experimental observations. In fact, in our samples, the substrate suppresses the lattice distortion, and the movements of the atoms are weaker than those in a free-standing twisted bilayer graphene[38]. Moreover, the in-plane deformation has a dominant effect on the electronic properties, which can be seen from the LDOS in Fig. 3e. In the deformed sample in Fig. 1c, the fitting parameters are the sets B (in-plane) and E (out-of-plane) labeled in Fig. 3.

The structure deformation leads to an electrostatic potential $V$, proportional to the local compression/dilatation, and a pseudo-vector potential **A**, associated with the shear deformation at one site. They can be written as[2,3]:

$$V = 2g_1 \frac{(d_1 + d_2 + d_3)/3 - d}{d},$$
$$A_x = c\frac{\beta\gamma_0}{d}(u_{xx} - u_{yy}), \qquad (2)$$
$$A_y = -c\frac{2\beta\gamma_0}{d}u_{xy},$$

where $g_1 \approx 4$ eV is the deformation potential for graphene, $d_1$, $d_2$, and $d_3$ are the first neighbor inter-atomic distances in the

deformed lattice, $d = 1.42$ Å is the equilibrium carbon–carbon distance, $c \approx 1$ is a numerical factor depending on the detailed model of chemical bonding, $\beta \approx 2$ is the electron Grüneisen parameter, $\gamma_0 = 3.2$ eV is the strength of first-neighbor interaction in the plane, $u_{ij}$ is the deformation tensor. The PMF is given by $\mathbf{B} = \nabla \times \mathbf{A}$, which can be estimated as[3]:

$$|\mathbf{B}| \approx \frac{\hbar c}{e}\left(\frac{2\beta}{3}\right)\frac{\overline{u}}{a_m d}, \qquad (3)$$

where $a_m$ is the period of the moiré pattern. And $\overline{u}$ is the magnitude of the shear deformation, which can be calculated from the first neighbor interatomic distances in the deformed sample as:

$$\overline{u} = \frac{\sqrt{3(d_3 - d_2)^2 + (d_2 + d_3 - 2d_1)^2}}{2d} \qquad (4)$$

The local deformation potential $V$ and the magnitude of the pseudo-magnetic fields $\mathbf{B}$ are plotted in Fig. 1d, e, respectively. Such structural deformation results in a non-uniform $\mathbf{B}$ with a maximum value of about 6 T. One of the effects of applying a strong external magnetic field in a two-dimensional electronic structure is the Landau quantization of the eigenstates and, subsequently, the quantum Hall effect in the transport properties. In our samples, the PMFs are not uniform, but form a ring-structured vortex lattice around the centers of the AA regions, similar to an Abrikosov vortex in superconductors[3]. Here, these pseudo-Landau levels are strongly correlated to the space-dependent PMFs, similar to the Landau levels that appear in

the presence of a real magnetic field. It is important to emphasize that the PMFs in low-angle TBG intrinsically arise from the lattice deformation. We expect that, as the displacements of the atoms are more significant in a structure with smaller twist angle[3], the induced PMFs will also be stronger.

**Twist angle and heterostrain effetcs.** The pseudo-Landau level behavior with twisted angle of 0.65° has also been studied by STM/S, in which the observed pseudo-magnetic fields are fitted to be 8 T (Supplementary Note 7). The calculated PMFs are in qualitative agreement with the fitted one from the $E_N$ formula. The fitted PMFs decrease more smoothly in the space comparing to the calculated values, and this difference can be explained in the following: (i) The expression of the Landau energies $E_N$ is valid for the case of bilayer graphene under a uniform external magnetic field. On the contrary, in our sample, the deformation-induced PMFs are non-uniform. (ii) As can be seen from the Landau energies $E_N$, the fitted PMFs can be modulated by the effective mass $m^*$, of which the value is taken from other experimental results[24,33], and the effective mass can also be space-dependent due to the local stacking structure. All in all, in this paper, both the theoretical and experimental values of maximum PMFs increase when the twist angle decreases, which is consistent with the fact that the strength of the lattice distortion becomes stronger for a TBG with smaller rotation angles.

The results presented so far have demonstrated that the twist angle can be an effective tuning parameter in tailoring the lattice deformation and subsequently tuning the electronic and magnetic properties in low dimensional systems. Next, we investigate heterostrain effects on the PMFs and their distributions in the low-angle TBG entity. The heterostrain is uniaxial, which originates from the pinning of the graphene layer at its grain boundaries or step edges during the epitaxial growth (Supplementary Note 4 and Note 9)[20]. Figure 4a shows the topography image with severely distorted moiré pattern. Such distorted moiré structures are due to the existence of uniaxial heterostrain, where

the moiré pattern can serve as a magnifying glass[20,28,29,39]. Namely, when the honeycomb lattice is slightly strained, the resulting moiré pattern is significantly distorted, in which the twist angle and strain can be determined by the STM images. For the sample in Fig. 4a, we obtain the twist angle and heterostrain to be $\theta = 0.98°$ and $\sigma = 0.78\%$, respectively (Supplementary Note 5).

Figure 4c, d shows dI/dV spectra and calculated LDOS curves taken along a line from an AA to an AB region. Similar to the findings in the $\theta = 0.48°$ sample, pseudo-Landau levels are also observed as a series of pronounced peaks. The energy spacings between the pseudo-Landau levels vary from the AA to AB region, which indicates the non-uniform characteristics of the PMFs. Significantly, the two low-energy pseudo-Landau levels near the Fermi level are similar to the correlation-induced splitting of flat bands reported recently in STM/S studies of the magic-angle TBG[16–18]. Such enhanced electronic correlation effects can be understood within the context of heterostrain in low-angle TBG (Supplementary Note 10)[20]. Figure 4b shows the corresponding linear fitting of pseudo-Landau levels, which gives a PMF value of 6 T. By linear extrapolating the LDOS at high energy in Fig. 4c, we deduced the center of the states, which changes with the positions. For our sample with a considerable heterostrain of $\sigma = 0.78\%$, it is still hard to visualize the atomic displacements from the high resolution STM image. Despite this, the topography image of large area moiré patterns shows distorted six-fold symmetry, which is in good agreement with our atomic simulations (Supplementary Fig. 5).

In accordance with recent reports[20,25], high energy peaks appear in AB regions, as shown in Fig. 4c, d. These peaks, associated with a partial band gap opening in a high energy moiré band[20,40] or localized domain wall modes arising from strongly confined moiré potentials[25], suffer minor changes in the presence of heterostrain and atomic deformations (Supplementary Note 8 for the effects of heterostrain and deformation on the electronic properties of TBG). This is expected since these lattice deformations and pseudo-magnetic fields only occur around AA regions,

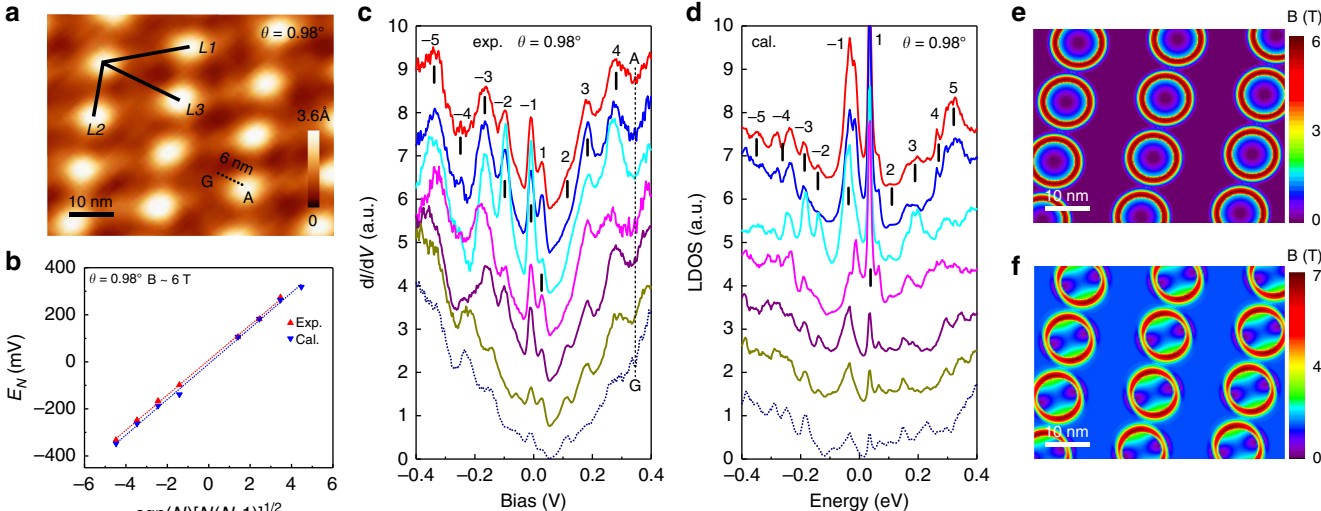

**Fig. 4 Electronic properties and PMF behaviors of TBG with uniaxial heterostrain. a** STM topography image (65 nm × 50 nm) with distorted moiré pattern. Twist angle $\theta = 0.98°$ with heterostrain $\sigma = 0.78\%$. Sample bias $V = 200$ mV, tunneling current $I_t = 1.0$ nA. The three moiré wavelengths are: $L1 = 20.3$ nm, $L2 = 12.3$ nm, and $L3 = 19.2$ nm. **b** Linear fittings of the peak energies and Landau index obtained from both the measured and calculated datas. The estimated PMF value is about 6 T. **c** dI/dV spectra with interval of 1 nm taken from AA to AB region indicated by black dots in **a**. The curves have been vertically shifted for clarity. **d** Local density of states computed from the same locations in **a**. The obtained features are in good agreement with our experimental results. **e, f** Calculations of the magnitude of the pseudo-manetic fields **B** produced by the atomic distortion of the top layer (unstrained one, **e**) and the bottom layer (strained one, **f**), respectively.

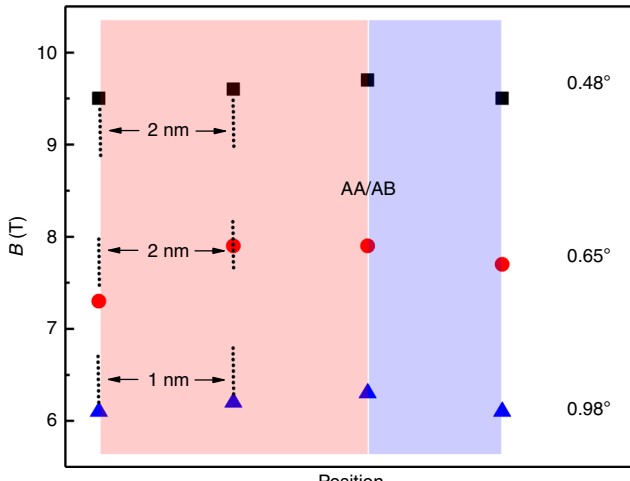

**Fig. 5 The fitted pseudo-magnetic field of TBGs with different twisted angles around the region of AA/AB transtion.** The obtained PMFs increase with the decreasing twisted angles and the PMF areas are distributed near the AA regions with its maximum value occuring at the AA/AB transitions, which is highly consistent with our calculated results.

as shown in Fig. 4e, f. Due to the considerable heterostrain applied to the bottom layer here, the shape and amplitude of the arising PMFs in each layer are no longer identical. Obviously, the strained layer has been more extended than the unstrained one, resulting in larger PMFs, which go up to 7 T. This tendency is in good agreement with previous findings[3,4]. Besides, the PMFs are stronger along the direction of the uniaxial heterostrain within each circular regions. As a consequence of uniaxial strain, the local deformation potential $V$ on the strained layer is 0.035 eV higher than that of unstrained one (Supplementary Note 9). Thereby, both the rotation angle and uniaxial heterostrain can be utilized to fine-tune the geometry, amplitude, and periodicity of the PMFs. In Fig. 5, we plot the twisted angle dependent PMFs and their spatial distributions around the AA regions. The locations in experiments are equally taken with interval of 1 nm or 2 nm. The experimental fitted results show that the PMFs increase with the decreasing twist angles and the PMF area occurs near the AA regions which reaches its maximum at the AA/AB transitions. The experiments and calculations show good agreements.

## Discussion

Collectively, the central findings achieved here may have important far-reaching implications. First of all, the existence of PMFs in TBG systems may finally offer testing grounds for observing quantum anomalous Hall effects as originally proposed[26]. Secondly, such PMFs are opposite for electrons at the K and K' valleys, leading to clockwise or anticlockwise valley vortex currents around the AA regions, which can be exploited for establishing quantum valley Hall effects[1]. Thirdly, the present study has demonstrated that the atomic deformations play a critical role in determining the electronic properties of the low-angle TBG, which in turn should also influence the emergent physics of correlated electrons[13–19]. Furthermore, similar physical phenomena, including the appearance of PMFs, may also be present in other van der Waals systems (for example, transition metal dichalcogenides bilayers or heterostructures[41–43]), and their effects on the electronic properties remain to be fully explored.

## Methods

**Sample fabrication and characterization**. The bilayer graphene samples were prepared by chemical vapor deposition on Cu–Ni alloy substrates using $CH_4$ as the carbon feedstock and $H_2$ as the reduction gas. Graphene growth was achieved at

1040 °C for 10–30 min after substrate was annealed for 30 min at 1000 °C under vacuum. The as-grown sample was then annealed in ultra-high vacuum at around 600 °C for several hours. The STM and STS measurements were conducted in a home-built STM with RHK R9 controller and RHK Pan-style scan head at 77 K with base pressure below $1.0 \times 10^{-10}$ Torr. The scanning tips were obtained by electrochemical etching of tungsten wires. STS measurements were carried out with standard lock-in technique with bias modulation of 5 mV at 2023 Hz.

**Tight-binding calculation**. The full tight-binding model by considering $p_z$ orbitals of all carbon atoms in twisted bilayer graphene is used to compute the local density states and other electronic properties of all the presented calculations. The model is developed in[20], and has been shown to reproduce accurately the differential conductance dI/dV measured from low-angel twisted bilayer graphene. In this model, the hopping integral $t_{ij}$ between $p_z$ orbitals at site $i$ and $j$ is described by:

$$t_{ij} = n^2 V_{pp\sigma}(r_{ij}) + (1 - n^2) V_{pp\pi}(r_{ij}),$$

where $r_{ij}$ is the distance between site $i$ and $j$, $n$ is the direction cosine along the direction perpendicular to the graphene layer $\boldsymbol{e_z}$. The Slater and Koster parameters $V_{pp\sigma}$ and $V_{pp\pi}$ follow

$$V_{pp\pi}(r_{ij}) = -\gamma_0 e^{q_\pi(1-r_{ij}/d)} F_c(r_{ij}),$$
$$V_{pp\sigma}(r_{ij}) = \gamma_1 e^{q_\sigma(1-r_{ij}/h)} F_c(r_{ij}),$$

where $d = 1.42$ Å is the nearest carbon-carbon distance, $h = 3.349$ Å is the interlayer distance, $\gamma_0$ and $\gamma_1$ were chosen to be 3.12 and 0.48 eV, respectively. $q_\sigma$ and $q_\pi$ satisfy $\frac{q_\pi}{d} = \frac{q_\sigma}{h} = 2.218$ Å$^{-1}$, and $F_c$ is a smooth function $F_c(r) = (1 + e^{(r-r_c)/l_c})^{-1}$, in which $l_c$ and $r_c$ were chosen to be 0.265 and 5.0 Å, respectively. The local density of states were calculated by using the recursion method in real space based on Lanczos algorithm[34]. To reach a precision in energy of 3.5 meV, we have computed 10,000 steps in the continued fraction built in a super-cell of 15 millions of orbitals with periodic boundary conditions.

## Data availability

The whole datasets are available from the corresponding author on reasonable request.

## Code availability

The codes used to construct the twisted bilayer graphene and calculate its electronic properties are available from the corresponding author on reasonable request.

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

## Acknowledgements

We thank Dr. Zhengfei Wang for insightful discussions. This work was supported by the financial support from the National Key Research and Development Program of China (Grant Nos. 2017YFA0205004, 2018FYA0305800, 2017YFA0303500), the National Natural Science Foundation of China (Grant Nos. 11474261, 11634011, 11774269, 61434002), the Fundamental Research Funds for the Central Universities (Grant Nos. WK3510000006 and WK3430000003), Anhui Initiative in Quantum Information Technologies (Grant No. AHY170000), and Strategic Priority Research Program of Chinese Academy of Sciences (Grant No. XDB30000000). Numerical calculations presented in this paper have been performed on a supercomputing system in the Supercomputing Center of Wuhan University.

## Author contributions

S.Y.Q., S.J.Y. and Z.Y.Z. co-supervised the project. H.H.S. and S.Y.Q. designed the experiments. H.H.S. mainly performed the STM/S experiments and data analysis with the assistance of R.X.Z., P.J.L. and K.X. Z.Z. mainly performed the theoretical calculations with the help from K.X.H., E.V.V. and J.A.S.G. Z.K.Q. and H.X.J. prepared graphene samples. H.H.S., Z.Z., M.I.K., S.J.Y., S.Y.Q. and Z.Y.Z. co-wrote the paper with input from the other authors.

## Competing interests

The authors declare no competing interests.
