## [Peer Review File · Nature Communications]

Reviewers' comments:

Reviewer #1 (Remarks to the Author):

The manuscript "Large-area, periodic, and tunable pseudo-magnetic fields in low-angle twisted bilayer graphene" by Shi et al. reports on the STM study of the tiny-angle twisted bilayer graphene. The authors report the experimental evidence of the existence of periodic pseudo-magnetic fields in twisted bilayer graphene. They also claim the geometry, amplitude, and periodicity of the pseudo-magnetic field can be finely tuned. The presented data are state-of-the-art and the manuscript is well organized. I have read the manuscript with great interest and identified several issues that make it impossible for me to recommend publication as it stands.

1. The authors claimed that "experimental explorations of strain engineering in the homostructural systems of TBG, especially the pseudo-magnetic behaviors, are still lacking". This is not true because that strain engineering in the TBG has been explored in experiment in literature, for example, in PRL 120, 156405 (2018), Nature Commun. 4, 2159 (2013), PRB 98, 235402 (2018), and so on. Some of these studies reported the pseudo-magnetic fields in the TBG.

2. The authors pointed out that a series of pronounced peaks with nearly equal energy spacing have been observed in a 0.48 degree TBG. Previously, similar spectrum has been reported in the first magic-angle TBG in STM measurement. However, such equal energy peaks are attributed to confined states of the moire pattern in the TBG. The authors should comment on this.

3. According to theory of this work, the pseudo-magnetic field in the TBG depends on the period of the moire pattern. To justify the observed peaks as pseudo-Landau levels, the authors should measure the pseudo-magnetic field in TBG with different twist angles and compare the experimental result quantitatively with that obtained in theory.

4. The pseudo-magnetic field does not break the time reversal symmetry of the system. To definitely demonstrate that the observed peaks as pseudo-Landau levels, the authors should measure the spectra by combining both the pseudo-magnetic fields and real magnetic fields. In such a case, it is expected to observe splitting of the peaks, as observed in strained graphene monolayer.

Reviewer #2 (Remarks to the Author):

Manuscript 211448 brings to light experimental evidence for artificial gauge fields generated by strain in twisted bilayer graphene (TBG), corroborated by tight-binding calculations. The authors point out interesting ways to control the magnitude and geometry of the associated pseudo-magnetic fields by changing the rotation angle or the strain in the material. This work reinforces the idea that twisted bilayer graphene is a very powerful platform for the manifestation of exotic phases of matter.

While the observation of pseudo-Landau levels in graphene platforms under strain have already been reported in several occasions, this work highlights the presence of artificial gauge fields in relaxed TBG, without external strain. The Referee finds these results relevant since these can have important implications for the correlated states and superconductivity observed in TBG at low temperatures.

This manuscript can be eligible for publication in Nature Communications once the authors clarify the points raised below.

a) Comparing the experimental and calculated dI/dV spectra in Fig. 1, we see many similarities, but also major discrepancies. First: the calculated spectra shows a very large peak at zero bias, while the experimental result shows a minor peak. Second: the calculated spectra show peaks at

higher voltages which are associated with the different Landau levels which increase as one goes to higher voltages (possibly associated with an increasing background DOS), but in the calculated scenario the peaks decrease at higher voltages. How can the authors explain these discrepancies?

b) A quick search on the literature shows that the observation of pseudo magnetic fields in mono and bilayer graphene has been reported earlier. In particular for twisted bilayer graphene:

Strain and curvature induced evolution of electronic band structures in twisted graphene bilayer
Wei Yan, Wen-Yu He, Zhao-Dong Chu, Mengxi Liu, Lan Meng, Rui-Fen Dou, Yanfeng Zhang, Zhongfan Liu, Jia-Cai Nie & Lin He
Nature Communications 4, 2159 (2013)

This reference is not cited by the authors. Both manuscripts show very similar results to this work. In particular the second reference Fig. 1e) is essentially the same as Fig. 1b) of the current manuscript.

c) Theoretical work on pseudo-magnetic fields in twisted bilayer graphene might also be worthy mentioning and relating to the presented work, in particular there are the following recent references:

Electrically Tunable Gauge Fields in Tiny-Angle Twisted Bilayer Graphene
Aline Ramires and Jose L. Lado
Phys. Rev. Lett. 121, 146801 (2018)

Pseudo Landau level representation of twisted bilayer graphene: Band topology and implications on the correlated insulating phase
Jianpeng Liu, Junwei Liu, and Xi Dai
Phys. Rev. B 99, 155415 (2019)

d) Could the authors be more precise about the symmetry of the lattice considering relaxation? What is the associated point group symmetry?

e) The Referee does not see the importance of the comparison of the structure of the pseudo-magnetic fields with the Abrikosov vortex lattice in superconductors. Is there any fundamental reason for this comparison?

f) Could the authors define the quantities d and γ_0 after Eq. 2?

Reviewer #3 (Remarks to the Author):

H. Shi and co-workers report an interesting STM/STS study of a twisted bilayer graphene on the Cu-Ni substrate. The authors demonstrate an existence of discrete Landau levels (LL) in strained bilayer graphene, attributed to the presence of the pseudomagnetic field. Considering the recent discoveries of unusual phenomena in bilayer graphene with extremely small twist angle, this work will be of fundamental interest to the research community in this field. However, the authors need to address the following questions before the recommendation for the publication in Nature communications.

- 1) Are these strained regions randomly distributed? How to control the strain in bilayer graphene?
- 2) How to extract the strain value for the bilayer graphene as shown in Figure 1 a and 3a? The authors need to provide the corresponding atomic models for these STM images.
- 3) How to determine the strain in each layer of bilayer graphene? The strong interaction with

underlying Cu-Ni substrate is expected to impose substantial structural distortion onto graphene lattice. The role of Cu-Ni substrate must be discussed more critically.

4) The calculated pseudo-magnetic field and local potential variation are presented in both Figure 1 and Figure 4. The author need to provide more experimental data to support these calculations. For example, Landau level Vs PMF, and Dirac Point VS local potential variation etc

5) The STS measurements constitute a core of the experimental evidence. The LL positions are known to depend on the square-root of both level index n and magnetic field B . The authors have to perform more thorough analysis of STS spectra: i) recover the linear dependence of E_n vs. $\text{sgn}(n)\sqrt{n}$, in order to discard the possibility of the tip-induced features, ii) estimate energy spacing of the LLs and extract effective intensity of the pseudomagnetic field (B_s) (see N. Levi, Science 329, 544 (2010)). The extracted B_s values need to be correlated with theoretically predicted maximum value of 6T (Figure 1d). Furthermore, the author needs to extract the map of the B_s , from spatially-acquired STS spectra, and compare that with the theoretically-calculated B map in Figure 1e.

6) Theoretically calculated B maps are presented by ring-like vortexes in the vicinity of the AA regions. The physic origin of these circular vortexes needs to be discussed in detail. The author may need to provide real space maps which show the lattice strain and deformation over ridges of these vortexes.

7) In Figure 1c, authors present calculated LODS spectra for twisted bilayer graphene with and without deformation. How was this "artificial" deformation included into theoretical modeling? It should be clearly specified.

Reviewer #1 (Remarks to the Author):

The manuscript "Large-area, periodic, and tunable pseudo-magnetic fields in low-angle twisted bilayer graphene" by Shi et al. reports on the STM study of the tiny-angle twisted bilayer graphene. The authors report the experimental evidence of the existence of periodic pseudo-magnetic fields in twisted bilayer graphene. They also claim the geometry, amplitude, and periodicity of the pseudo-magnetic field can be finely tuned. The presented data are state-of-the-art and the manuscript is well organized. I have read the manuscript with great interest and identified several issues that make it impossible for me to recommend publication as it stands.

1. The authors claimed that "experimental explorations of strain engineering in the homostructural systems of TBG, especially the pseudo-magnetic behaviors, are still lacking" . This is not true because that strain engineering in the TBG has been explored in experiment in literature, for example, in PRL 120, 156405 (2018), Nature Commun. 4, 2159 (2013), PRB 98, 235402 (2018), and so on. Some of these studies reported the pseudo-magnetic fields in the TBG.

Response: We thank the reviewer for the generally positive comments on this work.

We totally agree that the strain engineering of graphene layers has been extensively explored in recent years, both experimentally and theoretically. For example, the external strain engineering in monolayer and bilayer graphene, usually in the form of nanobubbles or wrinkles as displayed in Ref. [1-7, 20, 22-25] of the main text. Here, we want to emphasize that the observed pseudo-magnetic fields in this study arises from the lattice deformations in the strongly coupled graphene bilayers. In other words, the origin of the pseudo magnetic field is completely different from previous works, and our reported intrinsic pseudo-magnetic behaviors have not been experimentally studied yet.

The sentence in the main text mentioned above has been corrected as follows: "However, yet to date, experimental explorations of such intrinsic strain effects arising from lattice relaxations in the strongly coupled homostructural system, especially the pseudo-magnetic behaviors, are still lacking." We have also added "intrinsic" in the title to emphasize the unique contribution of our work. At the same time, the related papers have been cited in the main text as:

[24] Yan, W.; He, W.-Y.; Chu, Z.-D.; Liu, M.; Meng, L.; Dou, R.-F.; Zhang, Y.; Liu, Z.; Nie, J.-C.; He, L. Strain and Curvature Induced Evolution of Electronic Band Structures in Twisted Graphene Bilayer. Nat Commun 2013, 4, 2159.

[25] Qiao, J.-B.; Yin, L.-J.; He, L. Twisted Graphene Bilayer around the First Magic Angle Engineered by Heterostrain. Phys Rev B 2018, 98, 235402.

2. The authors pointed out that a series of pronounced peaks with nearly equal energy spacing have been observed in a 0.48 degree TBG. Previously, similar spectrum has been

reported in the first magic-angle TBG in STM measurement. However, such equal energy peaks are attributed to confined states of the moiré pattern in the TBG. The authors should comment on this.

Response: We thank the reviewer for the insightful suggestions.

Similar to earlier reports in the external strained bilayer graphene near magic angles, a series of equally spaced peaks exist in the AB stacking regions, which can be well explained as localized states by the partial band gap opening in higher energy moiré bands (Phys Rev Lett 120, 156405 (2018)) and localized domain wall modes arising from strongly confined moiré potentials (Phys Rev B 98, 235402 (2018)). However, in this work we show the similar peaks in and near the AA regions. In low-angle TBG, the AB stacking regions are nearly perfect Bernal stacked and are less affected by the lattice deformations and external strains. In contrast, the lattice distortion effects become significant considering that the interlayer interactions are strong near the AA regions, where the AA stacked atoms are not stable and tend to stack as AB-like mode. Our tight binding calculations show that the lattice deformation has to be considered in order to reproduce the pronounced peaks in this area.

This part of discussions can now be found in the second last paragraph of the main text as: "In accordance with recent reports [20, 25], high energy peaks appear in AB regions, as shown in Figs. 4c and 4d. These peaks, associated with a partial band gap opening in a high energy moiré bands [20, 40] or localized domain wall modes arising from strongly confined moiré potentials [25], suffer minor changes in the presence of heterostrain and atomic deformations."

[20] Huder, L.; Artaud, A.; Quang, T.; Laissardière, G.; Jansen, A.; Lapertot, G.; Chapelier, C.; Renard, V. Electronic Spectrum of Twisted Graphene Layers under Heterostrain. Phys Rev Lett 2018,120, 156405.

[25] Qiao, J.-B.; Yin, L.-J.; He, L. Twisted Graphene Bilayer around the First Magic Angle Engineered by Heterostrain. Phys Rev B 2018, 98, 235402.

[40] Wong, D.; Wang, Y.; Jung, J.; Pezzini, S.; DaSilva, A.; Tsai, H.-Z.; Jung, H.; Khajeh, R.; Kim, Y.; Lee, J.; et al. Local Spectroscopy of Moiré-Induced Electronic Structure in Gate-Tunable Twisted Bilayer Graphene. Phys Rev B 2015, 92, 155409.

3. According to theory of this work, the pseudo-magnetic field in the TBG depends on the period of the moiré pattern. To justify the observed peaks as pseudo-Landau levels, the authors should measure the pseudo-magnetic field in TBG with different twist angles and compare the experimental result quantitatively with that obtained in theory.

Response: We thank the reviewer for the important comments.

According to our theory, the pseudo-magnetic fields intrinsically arise from the lattice deformation in the strongly coupled low-angle TBG. With smaller twist angles the displacements of the atoms are more significant within each moiré pattern, leading to

stronger a pseudo-magnetic fields. In our revised manuscript, we compared several measured results with different twist angles below 1° (see Figure 1, Figure 4 in the main text, and Figure S7 in the Supplementary Information). As summarized in Fig. 5 in the revised main text, we extracted the fitted PMF value of 6 – 9 T while the twisted angle decreases from 0.98° to 0.48° . Such results are well explained by our theoretical models, and we further calculated the pseudo-magnetic fields with low-twisted angles (Figure 1e and Figure 4e,f). Our calculated results are in good agreement with our STM measurements overall, except that the measure PMF maximum is a slightly higher. For example, in the 0.48° sample, the calculated maximum PMF is 6 T while the fitted one is about 9 T. Such discrepancy can be explained with following reasons:

- a) The expressions of the Landau energies E_N is valid for the case of bilayer graphene under a uniform external magnetic field, and in such system, only the nearest-neighbor interactions are taken into account. On the contrary, in our sample, the moiré pattern contains different configurations, for instance, the AA and AB stackings, and the deformation-induced PMFs are not uniform. The LDOS is indeed not only determined by the PMFs at the point where it is measured, but also strongly affected by the PMFs nearby.
- b) As can be seen from the Landau energies E_N , the fitted PMFs can be modulated by the effective mass m^* , which is essentially space dependent. All in all, in this paper, the values of both calculated and fitted PMFs increase when the twisted angle decreases, which is consistent with the fact that the lattice distortion becomes stronger for a TBG with smaller rotation angle (Figure 5).

4. The pseudo-magnetic field does not break the time reversal symmetry of the system. To definitely demonstrate that the observed peaks as pseudo-Landau levels, the authors should measure the spectra by combining both the pseudo-magnetic fields and real magnetic fields. In such a case, it is expected to observe splitting of the peaks, as observed in strained graphene monolayer.

Response: We thank the reviewer for the interesting suggestions.

As explained in the answer to comment 3, the experimental observed peaks in LDOS around the region of AA/AB transition follow a dependence of $\sqrt{N(N-1)}$ where N is the index of the peak. This is consistent with the analytical expression of Landau levels in bilayer graphene ($E_N = \pm \hbar \omega_c \sqrt{N(N-1)}$, where N is the Landau index, \hbar is Planck's constant and $\omega_c = eB/m^*$, m^* is the effective mass (Phys Rev Lett **96**, 086805 (2006). Nat Phys **7**, 649 (2011)). And the estimated PMFs are about 9 T, 8T, and 6T for twisted angles of 0.48° , 0.65° , and 0.98° , respectively, consistent with the results extracted from the deformation. These are indeed extra evidences to prove the existence of pseudo-magnetic fields in measured samples. See also detailed discussions in the answers to previous comment.

Moreover, it has been well acknowledged that the PMFs in graphene induced by local strain engineering do not break the time reversal symmetry. But under external magnetic field, the

pseudo-Landau levels are splitted, such as strained monolayer graphene (ref. 35 in main text). We further calculated pseudo-Landau levels under external magnetic field (Figure 2b). We confirm that the existence of resonant peaks as pseudo-Landau levels and the splitting of pseudo-Landau levels are due to the break the time reversal symmetry of the deformed bilayer system. Such results are similar to the Landau levels splitting of strained graphene monolayer under external magnetic field in reference 35 of the main text.

[35] S. Y. Li, K. K. Bai, L. J. Yin, J. B. Qiao, W. X. Wang and L. He. Observation of unconventional splitting of Landau levels in trained graphene. Phys Rev B 92, 245302(2015).

Reviewer #2 (Remarks to the Author):

Manuscript 211448 brings to light experimental evidence for artificial gauge fields generated by strain in twisted bilayer graphene (TBG), corroborated by tight-binding calculations. The authors point out interesting ways to control the magnitude and geometry of the associated pseudo-magnetic fields by changing the rotation angle or the strain in the material. This work reinforces the idea that twisted bilayer graphene is a very powerful platform for the manifestation of exotic phases of matter.

While the observation of pseudo-Landau levels in graphene platforms under strain have already been reported in several occasions, this work highlights the presence of artificial gauge fields in relaxed TBG, without external strain. The Referee finds these results relevant since these can have important implications for the correlated states and superconductivity observed in TBG at low temperatures.

This manuscript can be eligible for publication in Nature Communications once the authors clarify the points raised below.

a) Comparing the experimental and calculated dI/dV spectra in Fig. 1, we see many similarities, but also major discrepancies. First: the calculated spectra shows a very large peak at zero bias, while the experimental result shows a minor peak. Second: the calculated spectra show peaks at higher voltages which are associated with the different Landau levels which increase as one goes to higher voltages (possibly associated with an increasing background DOS), but in the calculated scenario the peaks decrease at higher voltages. How can the authors explain these discrepancies?

Response: We thank the reviewer for the generally supportive comments.

First: In the scanning tunneling spectroscopy (STS) experiments, the measured dI/dV value is in arbitrary units, which is proportional to the local density of states of the tip position. In general, the intensities of the observed peaks are dependent on the scanning parameters, like tip-sample separation, set-point value, and sample bias etc. However, the peak positions in

energy are independent of these parameters. So, generally, it is more important to compare the position but not the magnitude of the Local density of states. Moreover, at the our experiment temperature ($\sim 80\text{K}$), the thermal broadening ($k_B T$) will also lead to the broadening of the resonant peaks, which in turn results in the decreased intensity of each peaks. To obtain the more distinct pseudo-Landau level peaks, we display the logarithmic dI/dV spectra (Figure 1b in the main text) and the background subtracted dI/dV spectra (Figure S3b in the Supplementary Information). These results show the well-defined Landau peaks and are well reproduced by our tight-binding calculations (Figure 1c in the main text). Additionally, we compare the experimental and calculated positions of pseudo-Landau level peaks as shown in Figure S3c in the Supplementary Information. Obviously, the calculated and measured results match well.

Second: We completely agree with the reviewer' s comments. The measured dI/dV values are associated with the increasing background DOS as the sweeping bias increases, that is, the tunneling current increases substantially at higher the sample bias. As shown in the fitted curve in Figure S3b in the Supplementary Information, the background-subtracted spectra show much more similarities with the calculated LDOS curve.

b) A quick search on the literature shows that the observation of pseudo magnetic fields in mono and bilayer graphene has been reported earlier. In particular for twisted bilayer graphene:

Strain and curvature induced evolution of electronic band structures in twisted graphene bilayer

Wei Yan, Wen-Yu He, Zhao-Dong Chu, Mengxi Liu, Lan Meng, Rui-Fen Dou, Yanfeng Zhang, Zhongfan Liu, Jia-Cai Nie & Lin He
Nature Communications 4, 2159 (2013)

This reference is not cited by the authors. Both manuscripts show very similar results to this work. In particular the second reference Fig. 1e) is essentially the same as Fig. 1b) of the current manuscript.

Response: We thank the reviewer' s helpful comments.

The related papers on strain engineering in bilayer graphene and recent STM studies on magic angle TBG have been cited in our revised manuscript. In the Nature Communications paper (Nat Commun 4, 2159 (2013)), the authors observed pronounced pseudo-Landau levels in the wrinkle of bilayer graphene. However, the origin of such pseudo-magnetic fields is quite different from ours. In their work, the PMFs originates from external strain engineering (in the form of wrinkles). In our work, we show that, in strongly coupled bilayer graphene, the lattice relaxations can induce strong **intrinsic** pseudo-magnetic fields in which the strength can be tuned by twisted angles. To make a more complete overview of related studies, the following references are added in our revised manuscript.

[16] Kerelsky, A.; McGilly, L.; Kennes, D.; Xian, L.; Yankowitz, M.; Chen, S.; Watanabe, K.; Taniguchi, T.; Hone, J.; Dean, C.; et al. Maximized Electron Interactions at the Magic Angle in Twisted Bilayer Graphene. *Nature* 2019, 572, 95–100.

[17] Xie, Y.; Lian, B.; Jäck, B.; Liu, X.; Chiu, C.-L.; Watanabe, K.; Taniguchi, T.; Bernevig, A.; Yazdani, A. Spectroscopic Signatures of Many-Body Correlations in Magic-Angle Twisted Bilayer Graphene. *Nature* 2019, 572, 101–105.

[18] Jiang, Y.; Lai, X.; Watanabe, K.; Taniguchi, T.; Haule, K.; Mao, J.; Andrei, E. Charge-Order and Broken Rotational Symmetry in Magic Angle Twisted Bilayer Graphene. *Nature* 2019, doi.org/10.1038/s41586-019-1460-4.

[19] Choi, Y.; Kemmer, J.; Peng, Y.; Thomson, A.; Arora, H.; Polski, R.; Zhang, Y.; Ren, H.; Alicea, J.; Refael, G.; et al. Electronic Correlations in Twisted Bilayer Graphene near the Magic Angle. *Nat Phys* 2019, 1–7.

[24] Yan, W.; He, W.-Y.; Chu, Z.-D.; Liu, M.; Meng, L.; Dou, R.-F.; Zhang, Y.; Liu, Z.; Nie, J.-C.; He, L. Strain and Curvature Induced Evolution of Electronic Band Structures in Twisted Graphene Bilayer. *Nat Commun* 2013, 4, 2159.

c) Theoretical work on pseudo-magnetic fields in twisted bilayer graphene might also be worth mentioning and relating to the presented work, in particular there are the following recent references:

Electrically Tunable Gauge Fields in Tiny-Angle Twisted Bilayer Graphene

Aline Ramires and Jose L. Lado

Phys. Rev. Lett. 121, 146801 (2018)

Pseudo Landau level representation of twisted bilayer graphene: Band topology and implications on the correlated insulating phase

Jianpeng Liu, Junwei Liu, and Xi Dai

Phys. Rev. B 99, 155415 (2019)

Response: We thank the reviewer for these suggestions.

The two theoretical work are now cited in our manuscript in the main text. The work (Phys Rev B 99, 155415(2019)) proposed that the low energy bands are equivalent to the pseudo Landau levels carrying two opposite Chern numbers in low-angle twisted bilayer graphene. The predicted pseudo Landau level descriptions are only valid for the AA stacking regions and would fail in the AB and BA regions. Such theoretical proposal is consistent with our experimental results.

The other work (Phys Rev Lett 121, 146801(2018)) also shows interesting physics in the low-angle TBG system where the electric field can be applied to tune the gauge fields of the sample. They analytically proposed that the emergence of pseudo Landau levels is associated with the emergent gauge fields. This is related but different from our works.

The following sentence is added in the first paragraph: "Such strain has been theoretically studied to significantly affected the electronic properties of the large-scale moiré pattern [22,23]."

d) Could the authors be more precise about the symmetry of the lattice considering relaxation? What is the associated point group symmetry?

Response: We thank the reviewer for the comments.

The lattice relaxation and associated point group symmetry are precisely explained in the revised main text as follows:

In this part, we assume that the deformed sample keeps the period of the rigid TBG. In the rigid sample with the AA point to be the rotation center, the moiré supperlattice has point group D_3 generated by a three-fold rotation about the axis formed by the AA points (C_3) and a two-fold rotation about the axis perpendicular to the AA point (C_2'). The deformed sample maintains the D_3 point group symmetry (Phys Rev B 99, 205134 (2019). Phys Rev B 96, 075311(2017)). For another sample with uniaxial heterostrain, the supperlattice for both rigidly twisted and deformed cases lose the D_3 symmetry.

e) The Referee does not see the importance of the comparison of the structure of the pseudo-magnetic fields with the Abrikosov vortex lattice in superconductors. Is there any fundamental reason for this comparison?

Response: We thank the reviewer's insightful comments.

Twisted bilayer graphene gives a very nice example of a system with tunable commensurability / incommensurability. Competition between in-layer elastic energy and inter-layer van der Waals interaction energy results in a reconstructed moiré pattern; its study is the main aim of our research. In one-dimensional case, starting from the seminal work [1], the incommensurability problem is discussed in terms of "soliton lattice". Very recently, the same concept was explicitly applied to the twisted bilayer graphene [2]. Initially, this concept was introduced to describe lattice mismatch which is relevant e.g. for graphene on hBN [3]. For the case of purely rotation-induced incommensurability, like in twisted bilayer graphene, it was hypothesized [4] that the atomic reconstruction can be described in terms of vortex lattice (and Abrikosov vortices in superconductors provide us a prototype example of such a lattice). We believe that this may be a fruitful idea to stimulate further theoretical and experimental research of the problem of two-dimensional commensurability, therefore we emphasized the analogy in our work.

[1] Frank F C, van der Merwe J H. One-dimensional dislocations. I. Static theory[J]. Proceedings of the Royal Society of London. Series A. Mathematical and Physical Sciences, 1949, 198(1053): 205-216.

[2] Ochoa H. Moiré-pattern fluctuations and electron-phason coupling in twisted bilayer

graphene[J]. arXiv preprint arXiv:1905.10850, 2019.

[3] Woods C R, Britnell L, Eckmann A, et al. Commensurate–incommensurate transition in graphene on hexagonal boron nitride[J]. Nat Phys, 2014, 10(6): 451.

[4] Van Wijk M M, Schuring A, Katsnelson M I, et al. Relaxation of moiré patterns for slightly misaligned identical lattices: graphene on graphite[J]. 2D Materials, 2015, 2(3): 034010.

f) Could the authors define the quantities d and γ_0 after Eq. 2?

Response: We thank the referee for his/her careful reading of our manuscript.

As clarified in the revised manuscript, $d = 1.42 \text{ \AA}$ is the equilibrium carbon-carbon distance, and $\gamma_0 = 3.2 \text{ eV}$ is the first-neighbours interaction in a plane.

Reviewer #3 (Remarks to the Author):

H. Shi and co-workers report an interesting STM/STS study of a twisted bilayer graphene on the Cu-Ni substrate. The authors demonstrate an existence of discrete Landau levels (LL) in strained bilayer graphene, attributed to the presence of the pseudo-magnetic field. Considering the recent discoveries of unusual phenomena in bilayer graphene with extremely small twist angle, this work will be of fundamental interest to the research community in this field. However, the authors need to address the following questions before the recommendation for the publication in Nature Communications.

1) Are these strained regions randomly distributed? How to control the strain in bilayer graphene?

Response: We thank the reviewer for the very supportive comments.

These strained regions are not randomly distributed, but form at the grain boundaries of the bilayer graphene or step edges of the CuNi alloyed surface during the epitaxial growth stage. For instance, the strained region in Figure 4a of the main text has a uniaxial heterostrain, in which the top layer graphene is unstrained and the bottom layer is slightly strained along one direction as we can derive from the STM image (that is, the deformed moiré pattern) and our atomic simulations. As displayed in Fig. S4a, the strained region with deformed moiré pattern in the right part meets with the unstrained region, where the grain boundary can be clearly observed. This result agrees well with previous studies of the origin of the heterostrain in the low-angle TBG (Phys Rev Lett 2018,120, 156405.). In this regard, we believe that the strain observed here are only the results of the arising grain boundaries and CuNi alloyed substrate in our CVD preparation method, where the pinning of the graphene layer at its boundaries results in slightly strained domains.

2) How to extract the strain value for the bilayer graphene as shown in Figure 1a and 3a? The

authors need to provide the corresponding atomic models for these STM images.

Response: We thank the reviewer very much for these suggestions.

Firstly, we need to clarify that the bilayer sample in Figure.1a in the main text is not strained (the strain value is zero) and for Figure. 4a the uniaxial heterostrain is 0.78% according to our calculations. The pseudo-magnetic field observed here originates from lattice reconstructions (the in-plane atomic displacements) of the strongly coupled graphene bilayers and the uniaxial heterostrain can be another degree of freedom to tune the strength and geometry of PMFs. Actually, the interlayer coupling of the graphene layers induced lattice deformations can be viewed as “intrinsic strain” , which substantially modifies the electronic properties of the system as previously studied (Phys Rev Lett 2012, 108, 216802.).

Then, for the value of the heterostrain displayed in Figure. 4a, detailed atomic models and STM images have been discussed in Section. E in the Supplementary Information. Generally, for twisted bilayer graphene, the observed moiré pattern in the STM image can be a magnifying glass for the determination of twisted angle and strain between the layers (Nano Lett 2017, 17, 2839-2843. Phys Rev Lett 2018,120, 156405. Nature 2019, 572, 95–100.). 1) For unstrained bilayers, the slightly mismatched honeycomb layers create a moiré superstructure with perfect hexagonal symmetry, as observed in Figure. 1a in the main text. The twisted angle can be exported by: $a_m = \sqrt{3}d/[2|\sin(\theta/2)|]$, where a_m is the moiré length and d is the carbon-carbon distance. 2) For the strained graphene lattice, the resulting moiré pattern loses its hexagonal symmetry and the pattern is significantly deformed as is shown in Figure. 4a in the main text and Figure S5 in Supplementary Information. Nevertheless, the strain value can be derived from the three moiré wavelengths by the STM measurements and theoretical atomic models. Details can be found in Supplementary Information Section E.

3) How to determine the strain in each layer of bilayer graphene? The strong interaction with underlying Cu-Ni substrate is expected to impose substantial structural distortion onto graphene lattice. The role of Cu-Ni substrate must be discussed more critically.

Response: We thank the reviewer' s comments.

Based on the deformed moiré pattern in the STM image, we have performed numerical analysis to determine the twist angle and strain strength. As is known, there exists two types of strain in the van der Waals bonded graphene bilayer. The first is homostain, where the uniaxial strain applied identically to both layers, and the resulting moiré pattern is only slightly deformed. Importantly, according to our tight binding calculations, the homostain has less effects on the electronic properties of the system, even for a large strain value, as is illustrated in Supplementary Information Section J. Another one is known as heterostrain where one of the graphene layer is strained along one direction, in which the moiré pattern is significantly affected by the uniaxial heterostrain. To this end, we performed atomic simulations together with calculations by defining the twist angle and strain value to

reproduce the as-observed moiré superstructure in the STM image (see details in Supplementary Information Section E), in which we obtained the heterostrain of about 0.78%. On the other hand, for a homo-strained bilayer system, the strain needs to be increased to 50% in order to reproduce the experimental data. Obviously, such large homostrain value is unreasonable in a graphene/metal system.

We totally agree with the reviewer that the CuNi alloyed substrate can impose impact onto the graphene both structurally and electronically. As has been previously reported, the Cu or Ni surface leads to electron doping, in which the doping level depends on the strength of interlayer coupling, and there exist moiré patterns due to the lattice mismatch of the substrate and graphene lattice (for example, graphene monolayer on Cu, *Nano Lett* 2010, 10, 3512–3516.). Here in the bilayer graphene/CuNi system, our measured dI/dV results (Figure S2) show that the bilayers are slightly electron-doped by the underlying CuNi substrate (the Dirac points were shifted about 20–64 meV for different twisted angles), indicating that the substrate has negligible effects on the electronic properties of the twisted bilayer graphene. On the other hand, theoretical studies of graphene/metal interface shows that the separation distance is directly correlated to the changes in the electronic structure of graphene (Figure S4b). An absolute binding energy of less than 2 eV can be considered as weak interaction at the graphene/metal interface (*Surf Sci Rep* 67, 83–115 (2012)). Therefore, for the Cu atom dominated CuNi(111) surface, it is logical to conclude that the alloyed surface is weakly bonded with the graphene layers. Thus, the PMFs behavior in this study must originate from the twisted bilayer graphene itself. However, from our simulation, it is clear that the substrate suppresses the atomic deformation of the twisted bilayer graphene.

Additionally, we have also performed calculations by relaxing the graphene islands on the Cu(111) surface as shown in Figure S4c in Supplementary Information Section D. It turns out that the graphene has a dome-like structure, where the central carbon atoms are 0.276 nm from the surface. The interlayer interactions are rather weak between the central C atoms and the substrate. However, the C atoms at the periphery is strongly bonded with the metal atoms, forming a *sp³-like* hybridizations. Such results are highly consistent with earlier reports (*Phys Rev Lett* 2012, 109, 265507). Thus, we suggest that strongly bonded graphene edge (that is, the grain boundary) provides extra driving force to form uniaxial heterostrain, and this point has been discussed above.

4) The calculated pseudo-magnetic field and local potential variation are presented in both Figure 1 and Figure 4. The author need to provide more experimental data to support these calculations. For example, Landau level Vs PMF, and Dirac Point VS local potential variation etc.

Response: We thank the reviewer' s comments.

The experimental observed peaks in LDOS around the region of AA/AB transition follow a dependence of $\sqrt{N(N-1)}$ where N is the index of the peak. This is consistent with the

analytical expression of Landau levels ($E_N = \pm \hbar \omega_c \sqrt{N(N-1)}$, where N is the Landau index, \hbar is Planck's constant and $\omega_c = eB/m^*$, m^* is the effective mass (Phys Rev Lett **96**, 086805 (2006). Nat Phys **7**, 649 (2011)) in bilayer graphene, and the estimated PMFs are about 9 T, 8T, and 6T for twisted angles of 0.48°, 0.65°, and 0.98°, respectively, agree well with the results extracted from the deformation. These are indeed extra evidences to prove the existence of pseudo-magnetic fields in measured samples. Detailed discussion and explanation are added in the main text of the new manuscript, with new Figure 2, Figure 4, Figure 5 and Figure S7. The variation of the Landau level can be supported by the LDOS in Fig. 4c where the energy spacing between the LL vary from the AA to AB regions, which gives a variation of PMF plotted in the new Fig. 5 in the main text. By linear extrapolating the LDOS in Fig.4c, we obtained the Dirac point for each point, as shown in Figure R1, which also changes with the positions. Both the PMF and Dirac point reach the maximum value at position C, which is

agreement with our calculations.

Figure R1 The energy of Dirac point changes with the position as illustrated in the Fig. 4a in the main text. The Dirac points are deduced by linear extrapolating the measured LDOS in Fig.4c.

5) The STS measurements constitute a core of the experimental evidence. The LL positions are known to depend on the square-root of both level index n and magnetic field B . The authors have to perform more thorough analysis of STS spectra: i) recover the linear dependence of E_n vs. $\text{sgn}(n)\sqrt{n}$, in order to discard the possibility of the tip-induced features, ii) estimate energy spacing of the LLs and extract effective intensity of the pseudo magnetic field (B_s) (see N. Levi, Science 329, 544 (2010)). The extracted B_s values need to be correlated with theoretically predicted maximum value of 6T (Figure 1d). Furthermore, the author needs to extract the map of the B_s , from spatially-acquired STS spectra, and compare that with the theoretically-calculated B map in Figure 1e.

Response: We thank the reviewer's insightful suggestions.

We completely agree with the reviewer's comments about the pseudo-Landau level positions and pseudo-magnetic field relations. For the bilayer graphene under the external magnetic field, the massive Dirac fermions are quantized with energies $E_N = \pm \hbar \omega_c \sqrt{N(N-1)}$.

In the absence of both an applied electric field and interactions, the $N = 0$ and $N = 1$ LLs are further degenerate, with an eight-fold degeneracy occurring at 0 meV. In our revised version of this manuscript, by fitting the equation above, we confirm the linear relationship of the Landau index and Landau peaks with different twisted angles (see Figure 2a, Figure 4b, and Figure S7d), which confirms the existence of pseudo-Landau levels in our observations. As summarized in Fig. 5 in the revised main text, we extracted the fitted PMF value of 6 – 9 T while the twisted angle decreases from 0.98° to 0.48° . Such results are well explained by our theoretical models, and we further calculated the pseudo-magnetic fields with low-twisted angles (Figure 1e and Figure 4e, 4f). Our calculated results are in good agreement with our STM measurements overall, except that the measure PMF maximum is a slightly higher. For example, in the 0.48° sample, the calculated maximum PMF is 6 T while the fitted one is about 9 T. Such discrepancy can be explained with following reasons:

- a) The expressions of the Landau energies E_N is valid for the case of Bernal stacking bilayer graphene under a uniform external magnetic field, and in such system, only the nearest-neighbor interactions are taken into account. On the contrary, in our sample, the moiré pattern contains different configurations, for instance, the AA and AB stackings, and the deformation-induced PMFs are not uniform. The LDOS is indeed not only determined by the PMFs at the point where it is measured, but also strongly affected by the PMFs nearby.
- b) As can be seen from the Landau energies E_N , the fitted PMFs can be modulated by the effective mass m^* , which is essentially space dependent. All in all, in this paper, the values of both calculated and fitted PMFs increase when the twisted angle decreases, which is consistent with the fact that the lattice distortion becomes stronger for a TBG with smaller rotation angle (Figure 5).

6) Theoretically calculated B maps are presented by ring-like vortexes in the vicinity of the AA regions. The physic origin of these circular vortexes needs to be discussed in detail. The author may need to provide real space maps which show the lattice strain and deformation over ridges of these vortexes.

Response: We again appreciate this comment.

As we discussed in the paragraph before Eq. (1) in the main text, the physical origin of the ring-like vortex is the interplay between the interlayer interaction and the in-plane strain fields. We extended the discussion of the origin in the same paragraph and include two references (2D Mater **2**, 034010 (2015). 2D Mater **5**, 015019 (2018)), which have a deep discussion about the atomic deformation in twisted bilayer graphene.

“That is, on one hand, the in-plane forces move atoms to maximize the area of AB/BA stacking domains, which has the minimum binding energy. On the other hand, the strain induced by the atomic displacements hinders such in-plane atomic rearrangement.”

From Eq. (2) and (4), the local deformation potential V in Figure 1 d in the main text indicates the map of the local compression/dilatation, and the pseudo-magnetic field in

Fig.1e illustrates the map of the shear deformation. Apart from these results, we also added the maps of the in-plane displacements $|\Delta d|$ and out-of-plane displacements $|\Delta z|$ for twisted bilayer graphene with twist angle $\theta = 0.48^\circ$ in Fig. 3 c and d in the main text and Fig. S6 c and d for twisted bilayer graphene with $\theta = 0.98^\circ$ in the Supplementary Information, respectively. All these maps of real movements are in agreement with the results from the molecular dynamics simulations in the reference (2D Mater 5, 015019 (2018)).

7) In Figure 1c, authors present calculated LODS spectra for twisted bilayer graphene with and without deformation. How was this “artificial” deformation included into theoretical modeling? It should be clearly specified.

Response: We thank the reviewer’s comments.

As has been stressed in the revised main text, the deformed samples are constructed in the following procedure:

“The deformed samples are constructed in the following procedure: First, the rigidly twisted bilayer graphene is constructed according to the commensurability and then the position of each atom is well defined. Next, we modify the in-plane and out-of-plane positions of atoms according to the position-dependent expression in Eq. (1) of the main text. For instance, for the in-plane deformation, individual atom move Δd towards the nearest AA point. In the out-of-plane deformation, the individual atom has a displacement in the z direction with + sign for the top layer and – sign for the bottom layer. The maps of ΔD and ΔZ for TBG with 0.48° are plotted in Fig. 3c and 3d, respectively. The movements of the atoms have the same tendency as that obtained from the molecular dynamics simulations. In this part, we assume that the deformed sample keeps the period of the rigid TBG. ”

Reviewers' comments:

Reviewer #1 (Remarks to the Author):

I have carefully studied the author's rebuttal and revisions made. The authors have done a remarkably good job for unambiguous verification of their claims using various experimental and computational tools. However, there are still three important issues (two large discrepancies between experiment and theory) that the authors should address. After that, I can make my final decision.

1. The authors studied carefully the electronic structures of the 0.48° TBG in Fig. 1. Recent experiment demonstrated that there should be structure reconstruction in such a TBG (Nature Materials.18 448 (2019)). And according to the theoretical calculation with considering the structure reconstruction, the flat bands at the charge neutrality point of this twist angle (very close to the second magic angle) should be strongly destroyed, which consists well with the experimental result of this work (Fig. 1b), where the peak at the charge neutrality in the LDOS is rather weak. However, the theoretical result still show rather flat bands at the charge neutrality point and the peak at the charge neutrality in the LDOS is the strongest. The authors should discuss the discrepancy.

2. The authors calculated the LDOS of the TBG in the presence of both pseudomagnetic fields and real magnetic fields, as shown in Fig. 2. Unexpected, the obtained valley splitting is extremely small. According to the result in Fig. 2, the valley splitting is only 15 meV when the pseudomagnetic field is 9 T and the real magnetic field is 10 T. However, the expected valley splitting should be about one order larger according to the formula of the Landau quantization in bilayer graphene. Additionally, the valley splitting is almost zero when the pseudomagnetic field is 9 T and the real magnetic field is 6 T. In such a case, the expected valley splitting should be larger than 100 meV. Does such a large discrepancy indicate that the peaks in the LDOS of the TBG should not be treated as pseudo-Landau levels?

Minor point: The authors reported two peak flanking the Fermi levels in the 0.98° TBG (which is very close to the first magic angle 1.05°). Such a result is quite similar as the correlation-induced splitting of the flat bands, as reported very recently in the magic-angle TBG (Nature572, 101 (2019), Nature572, 95 (2019), Nature572, 91 (2019)). The authors should discuss this.

Reviewer #2 (Remarks to the Author):

I have read the author's responses to the questions raised by me and by the other referees. I believe the authors made substantial improvements in the manuscript and clarified important points in the discussion. I recommend the manuscript for publication.

Reviewer #3 (Remarks to the Author):

the authors have addressed my questions. I recommend the publication of this work in Nature Communications

Responses to Reviewer #1's Remaining Comments (NCOMMS-19-17354)

We thank all the three reviewers for their further review of this manuscript. We are pleased that all the three reviewers are quite positive about this work for publication in Nature Communications. Specifically, Reviewers #2 and #3 have recommended acceptance, while Reviewer #1 raised some remaining issues to be further clarified before potential acceptance. As detailed below, Reviewer #1's constructive comments have helped us to further improve the paper. In particular, his/her comments prompted us to perform additional calculations to better interpret the experimentally observed suppression of the flat bands near the second magic angle, reported in the literature most recently (Ref. 15). Furthermore, we have expanded our discussions on the magnitudes of the valley splittings induced by the concerted effects of both the pseudo-magnetic field and external magnetic field, especially the important role of the non-uniform pseudo-magnetic field. With these improvements, we hope the paper is now acceptable for publication in Nature Communications.

Comment 1. *The authors studied carefully the electronic structures of the 0.48° TBG in Fig. 1. Recent experiment demonstrated that there should be structure reconstruction in such a TBG (Nature Materials.18 448 (2019)). And according to the theoretical calculation with considering the structure reconstruction, the flat bands at the charge neutrality point of this twist angle (very close to the second magic angle) should be strongly destroyed, which consists well with the experimental result of this work (Fig. 1b), where the peak at the charge neutrality in the LDOS is rather weak. However, the theoretical result still show rather flat bands at the charge neutrality point and the peak at the charge neutrality in the LDOS is the strongest. The authors should discuss the discrepancy.*

Response: We thank the reviewer for raising this critical point, which prompted us to have a closer comparison between the results presented here and in that Nature Materials paper (Ref. 15). This effort convinced us that there is actually no discrepancy between our theoretical results and those shown in Ref. 15. What we showed for the 0.48° TBG is the *local* density of states (LDOS) around the AA and AB regions (Fig. 1c here); in contrast, Ref. 15 displayed the *total* density of states (DOS) (Fig. 2c there). To have a more direct comparison, we have performed additional calculations of the total DOS, and the results are qualitatively very similar to those in Ref. 15, i.e., the flat bands near the second magic angle (0.47°) are indeed strongly suppressed due to the structural reconstruction (see Fig. R2).

Fig. R2. Calculated total density of states of 0.48° TBG with and without lattice deformation.

Comment 2. *The authors calculated the LDOS of the TBG in the presence of both pseudo-magnetic fields and real magnetic fields, as shown in Fig. 2. Unexpected, the obtained valley splitting is extremely small. According to the result in Fig. 2, the valley splitting is only 15 meV when the pseudo-magnetic field is 9 T and the real magnetic field is 10 T. However, the expected valley splitting should be about one order larger according to the formula of the Landau quantization in bilayer graphene. Additionally, the valley splitting is almost zero when the pseudo-magnetic field is 9 T and the real magnetic field is 6 T. In such a case, the expected valley splitting should be larger than 100 meV. Does such a large discrepancy indicate that the peaks in the LDOS of the TBG should not be treated as pseudo-Landau levels?*

Response: In the strained graphene samples, the magnitudes of valley splitting depend on both the pseudo-magnetic field and external magnetic field, as discussed before (Ref. 35). The theoretically expected valley splitting is valid only for well-defined K and K' valleys, where the pseudo-magnetic field is uniform. In these regions, the translational symmetry is preserved. However, in our systems the observed pseudo-magnetic field has drastically varying ring-like structures in real space. As a result, the two valleys are not well defined; therefore, the theoretically expected valley splitting value is no longer applicable. Instead, the valley splitting observed in this study (Fig. 2b) is extrapolated from the regions where the pseudo-magnetic field only varies mildly in real space.

In our revised manuscript, we have added some more discussions to clarify on this issue (page 3):

“Similar valley splitting phenomena have been reported in earlier studies of strained monolayer graphene [35], except that in the present study, such valley splitting is not as large as theoretically expected. The suppression can be understood, because the theoretically expected valley splitting is valid only for well-defined K and K' valleys, where the PMF is uniform. While in our systems the observed PMF has drastically varying ring-like structures in real space. As a result, the two valleys are not well defined; therefore, the theoretically expected valley splitting value is no longer applicable. Instead, the valley splitting observed in this study is extrapolated from the regions where the pseudo-magnetic field only varies mildly in real space.”

Minor point. *The authors reported two peak flanking the Fermi levels in the 0.98° TBG (which is very close to the first magic angle 1.05°). Such a result is quite similar as the correlation-induced splitting of the flat bands, as reported very recently in the magic-angle TBG (Nature 572, 101 (2019), Nature 572, 95 (2019), Nature 572, 91 (2019)). The authors should discuss this.*

Response: We appreciate the reviewer's suggestion on the potential broader significance of our findings in connection with the recent three reports in Nature. In the revised main text, we have added the following discussions on page 5:

“Significantly, the two low-energy pseudo-Landau levels near the Fermi level are similar to the correlation-induced splitting of the flat bands reported recently in STM/S studies of magic-angle TBG [16-18]. Such enhanced electronic correlation effects can be understood within the context of heterostrain in low-angle TBG [20].”

REVIEWERS' COMMENTS:

Reviewer #1 (Remarks to the Author):

The authors have well addressed my questions. Now I recommend the publication of this work.